# Tibiotarsal Arthrodesis with Retrograde Intramedullary Nail and RIA Graft: A Salvage Technique

**DOI:** 10.3390/jfmk8030122

**Published:** 2023-08-21

**Authors:** Giancarlo Salvo, Salvatore Bonfiglio, Marco Ganci, Salvo Milazzo, Rocco Ortuso, Giacomo Papotto, Gianfranco Longo

**Affiliations:** Department of Orthopedic Surgery, Trauma Center, Cannizzaro Hospital, 95100 Catania, Italymrcganci@gmail.com (M.G.);

**Keywords:** ankle arthrodesis, retrograde nail, infections, fractures

## Abstract

Ankle arthrodesis is a commonly used salvage procedure in the management of post-traumatic ankle fractures, which often result in severe disability and may require the amputation of the distal third of the leg. Successful ankle arthrodesis relies on a thorough assessment of local and systemic risk factors to ensure optimal results. Failure to accurately assess these factors may lead to unsatisfactory results. High-energy trauma causing bone defects and soft tissue necrosis often results in osteomyelitis, a condition that poses a significant threat to the success of the arthrodesis procedure. It is important to apply a standardised surgical protocol to minimise the possibility of superficial and deep infection and limit damage to the neighbouring soft tissues. Therefore, it is critical to undertake surgical lavage and debridement and administer systemic and local antibiotic therapy, along with the use of a spacer, to eradicate infection prior to performing arthrodesis. In this study, we present our experience in the recovery of limbs with post-traumatic complications via tibio-astragalic or tibio-calcaneal arthrodesis using a retrograde intramedullary nail technique. The approach involves a multi-step procedure using a previous antibiotic spacer implant and an autologous bone graft (RIA). This study spanned a period from January 2014 to December 2021 and included 35 patients (12 women and 23 men) with a mean age of 47.8 ± 20.08 years (range: 22–85 years). Among the patients, 18 had osteomyelitis following AO 43 C3 fractures, and 9 of them had previous exposure and bone loss at the time of injury. The remaining cases included 10 patients with AO 44 C fracture outcomes and 7 patients with AO 44 B fracture outcomes. Our results emphasise the importance of the meticulous management of local and systemic risk factors in ankle arthrodesis procedures. The successful eradication of infection and subsequent arthrodesis can be achieved via the implementation of surgical lavage, debridement, and systemic and local antibiotic therapy using spacers. This surgical protocol implemented by us has yielded excellent results, saving affected limbs from post-traumatic complications and avoiding the need for amputation. Our study contributes to the existing knowledge supporting the use of retrograde arthrodesis with intramedullary nails in severe cases where limb salvage is the primary goal. However, further research and long-term follow-up studies are needed to validate these results and evaluate the effectiveness of this technique in a larger patient population.

## 1. Introduction

Fractures of the distal tibia epiphysis can result in complex injuries with severe complications, particularly when accompanied by damage to the delicate soft tissue surrounding the distal tibia. These complications may include deformities, post-traumatic arthrosis, and osteomyelitis [1]. Infections affecting the distal third of the leg pose significant challenges for orthopaedic surgeons and are often associated with a high risk of limb amputation [2]. If the infection cannot be controlled initially and progresses to chronic osteomyelitis with infection of the ankle joint, reconstructive surgery with limb salvage must be performed for acute and chronic infection of the distal end of the tibia and ankle joint, but this surgery remains very challenging, despite all the surgical advances made in recent decades [3].

### 1.1. History of Literature

Ankle arthrodesis has emerged as a definitive treatment for various complications affecting the distal third of the leg. The operation was first described by Edward Albert in 1879 [4] and refined by John Charnley until internal fixation became a viable option, and, in 1983, Schneider described the arthroscopic arthrodesis of the ankle joint [5]. The approach to ankle arthrodesis is divided into open and arthroscopic techniques. The open approach can be performed using different, often combined surgical accesses. However, open arthrodesis is associated with higher rates of wound complications due to the large amount of soft tissue dissection required. Therefore, open approaches are generally reserved for patients with moderate to severe ankle deformities with healthy intact soft tissues [6]. Numerous techniques for ankle arthrodesis have been proposed in the medical literature, employing different fixation methods such as plates, screws, nails, and external fixators. Many surgeons in the literature prefer to use screw fixation as the main means of internal fixation because screws are easy to use, inexpensive, and require small surgical access. However, higher rates of pseudoarthrosis have been reported, especially in osteoporotic bone [7]. Others prefer plates over screws to achieve better consolidation rates, but the extensive dissection required to place the plate can lead to a higher risk of infection [8]. Arthrodesis with a retrograde intramedullary nail is the most widely used, but it determines the arthrodesis of the ankle and subastragalic joints. The subastragalic joint allows inversion and eversion and is essential for gait stability even when joint motion is reduced after arthrodesis. External fixation is generally indicated for complex patients with abundant bone defects and poor skin and soft tissue conditions, but consolidation rates are lower than internal fixation [7]. Among these techniques, retrograde intramedullary nail fixation for ankle joint fusion was first introduced by Gerhard Kuntcher in 1967, utilising the nail bearing his name [9]. Subsequent surgical approaches have been proposed, including the technique described by Kelikiam, which involved a lateral approach, fibula removal in the supra-syndesmotic region, and utilisation of the fibula as an autologous graft [10]. In recent years, there has been a shift towards less invasive surgical techniques, with ankle arthrodesis increasingly performed using percutaneous cannulated screws or closed retrograde nails. These approaches aim to minimise trauma to the surrounding tissues, which may have undergone previous skin grafting procedures.

### 1.2. Our Experience and Aims

In our experience, we have opted to forgo the use of the fibula as a bone graft in ankle arthrodesis procedures. Instead, we have favoured the utilisation of autologous grafts obtained using the retrograde intramedullary nail technique, employing the ipsilateral femur as the source. This study aims to share our successful experience with limb salvage using ankle arthrodesis in post-traumatic outcomes. Specifically, we have employed tibio-astragalic or tibiocalcaneal arthrodesis using the retrograde intramedullary nail technique. Prior to the arthrodesis procedure, it is imperative to address any existing infections, as high-energy trauma resulting in bone defects and soft tissue necrosis can lead to osteomyelitis, thus jeopardising the final outcome [11]. To ensure successful arthrodesis, surgical lavage and debridement, along with systemic and local antibiotic therapy, are required until the infection is eradicated. The use of antibiotic spacers aids in creating a favourable environment for the subsequent arthrodesis procedure. In this paper, we present the outcomes of our surgical approach, including bone callus consolidation, ankle arthrodesis, and functional recovery. We also discuss the complications encountered during the treatment process, along with their management. Furthermore, we assess the presence of pain near the apex of the retrograde nail and radiolucency as potential indicators of adverse outcomes. Lastly, we examine the association between consolidation indices and the lengths of the nails used in the procedures. We present positive results of limb salvage by sharing our experiences and outcomes. We aim to contribute to the existing knowledge and understanding of ankle arthrodesis as a salvage procedure for severe post-traumatic complications.

## 2. Materials and Methods

This retrospective study was conducted from January 2014 to December 2021 in our department Cannizzaro Hospital of Catania, involving 35 patients who underwent retrograde nail arthrodesis as a salvage method. The study population consisted of 12 women and 23 men, with an average age of 47.8 ± 20.08 years (range: 22–85 years). The patients included in the study presented with various complications resulting from post-traumatic osteomyelitis. Among the patients treated in our study, 2 (5.7%) patients had insulin-dependent type II diabetes mellitus and hypertension, 5 (14.2%) patients suffered from hypertension, 23 (65.7%) patients were smokers, and 5 (14.2%) patients had no other comorbidities. Among them, 18 (51.4%) patients had AO 43 C3 fractures, with 9 of these cases having previous exposure and bone loss at the time of injury. The remaining cases included 10 (28.5%) patients with AO 44 C fracture outcomes and 7 (20%) patients with AO 44 B fracture outcomes. Biopsy samples were obtained from all patients during surgical procedures, including interventions in the operating theatre, emergency room, and subsequent osteosynthesis, curettage, and debridement surgeries. The purpose of these biopsy samples was to analyze and identify the causative bacteria responsible for the infections. The laboratory analysis showed that the bacteria found to be responsible for the infections included Staphylococcus aureus in 16 (45.7%) patients, Staphylococcus aureus MRSA in 12 (34.2%) patients, Escherichia coli in 3 (8.5%) patients, and Acinetobacter baumani in 4 (11.4%) patients (Table 1).

This study focused on evaluating the outcomes of retrograde nail arthrodesis in this patient population. Data on bone callus consolidation, ankle arthrodesis, and functional recovery were collected and analyzed. The consolidation time, defined as the duration required for successful bone callus consolidation and consecutive ankle arthrodesis, was recorded for each patient.

Complications associated with the procedure were also documented. This included the occurrence of septic complications, such as surgical wound dehiscence, as well as other potential complications, such as injuries to adjacent muscles. Additionally, pain near the apex of the retrograde nail and signs of radiolucency were monitored as potential indicators of adverse outcomes. The lengths of the nails used in the procedures were also considered for their potential impact on consolidation indices.

Statistical analysis were performed using IBM SPSS Statistics Software for Macintosh to assess the significance of the results obtained. We used the Student *t*-test to compare the consolidation indices with nail lengths used and the healing time with respect to smokers and non-smokers with the aim of identifying any statistically significant differences. We set the sensitivity value for values below 0.05.

In conclusion, the Materials and Methods section provides an overview of the patient population, fracture types, and the microbiological analysis of biopsy samples. It also outlines the focus of this study, including the evaluation of outcomes such as bone callus consolidation and ankle arthrodesis. Furthermore, it mentions the documentation of complications and the statistical analysis conducted to assess the obtained results.

### Surgical Technique

The surgical technique for retrograde nail arthrodesis in our study involved a two-stage procedure to ensure optimal outcomes and promote the successful fusion of the ankle joint. Here, we provide a detailed description of each stage.

Stage 1:

Removal of implanted device: The first step involved the careful removal of any previously implanted devices from the affected ankle joint.

Surgical lavage: A thorough surgical lavage was performed to clean the macroscopically infected and skeletonised bone segments. This process aimed to remove any debris, necrotic tissue, or contaminants from the surgical site.

Biopsy and culture examination: Five biopsy samples were taken from the surgical site for culture examination and subsequent antibiogram. This allowed for the identification of the causative organisms and informed the selection of appropriate antibiotic therapy.

Antibiotic-treated cemented spacer implantation: To facilitate the formation of a biological cavity and promote surgical site remediation, an antibiotic-treated cemented spacer was implanted. The spacer consisted of a combination of 40 g of polymethyl methacrylate (PMMA) and 120 mg of gentamicin sulphate. This spacer not only acted as a temporary support but also helped eradicate the infection.

External fixator placement: An external fixator was implanted to stabilise the bone during the healing process of the surrounding soft tissues. This provided additional support and ensured proper alignment during the initial stages of recovery.

Stage 2:

Patient positioning: The second surgical stage was performed at a later time, usually 6 to 8 weeks after the initial procedure. The patient was placed in a supine decubitus position on a trauma bed, which facilitated the retrograde retrieval of the RIA from the ipsilateral femur. The knee was flexed to a maximum of 30° to allow easier access.

Transfibular approach: Arthrodesis was performed using the transfibular approach initially proposed by Adams and later modified by Wichman and Kelikian. This involved an anterolateral incision aligned with the 4th metatarsal, extending to the distal third of the peroneal diaphysis.

Fibula osteotomy: Approximately 2 cm from the articular rim, a fibula osteotomy was performed to remove the entire peroneal malleolus. This step provided better exposure of the tibio-tarsal and subastragalic joints, allowing for more accurate and reliable joint preparation.

Joint preparation and correction of deformities: After the cartilaginous lining was removed, any varo-valgus deformities of the tibio-ankle and pronation-supination of the sub-astragalic joint were corrected. The alignment of the heel on the leg was carefully checked to ensure proper joint realignment.

Removal of the antibiotic-cemented spacer: The antibiotic-cemented spacer was then removed, taking care not to damage the newly formed biological cavity. A biopsy specimen was taken for extemporaneous testing, and an additional five biopsies were collected. If the extemporaneous test yielded negative results, the intramedullary canal was reamed using increasing motorised burs to enhance the cruentation of the articular surfaces of the sub-astragalic and tibio-tarsal joints.

Placement of retrograde nail and RIA: Once the intramedullary canal was prepared, the retrograde nail and RIA were carefully placed within the newly formed biological cavity. Special attention was given to ensuring proper positioning and alignment of the components.

Closure and suturing: the biological cavity was meticulously sutured, securing the RIA in place. The overlying skin was also sutured to promote healing and prevent infection.

Immobilisation and postoperative loading: Following the completion of the surgical treatment, no immobilisation with a plaster cast or brace was applied. Gradual partial loading was permitted after 4 to 6 weeks, and full loading was allowed at 8 to 10 weeks post-surgery to facilitate the recovery process.

By following this two-stage surgical technique, we aimed to optimise the outcomes of retrograde nail arthrodesis for the treatment of severe ankle fractures. The approach prioritised infection control, joint realignment, and stability, ultimately leading to successful bone callus consolidation, ankle arthrodesis, and functional recovery in our patient population.

## 3. Results

In our study, we evaluated the outcomes of all patients who underwent retrograde nail arthrodesis for the treatment of severe ankle fractures. The results demonstrated successful bone callus consolidation and consecutive ankle arthrodesis in all patients. The average consolidation time was approximately 229.4 days ± 48.7 days, with a range between 181 and 277 days. This indicates that fusion of the ankle joint was achieved in a relatively short period of approximately 7.6 months after surgery. After the surgical procedure, patients were able to resume their daily work activities at an average of 257.5 ± 69.3 days after surgery. The range for this parameter was between 189 and 330 days, indicating a relatively quick recovery and return to normal functioning in approximately 8.5 months after the last surgical procedure. However, it is important to note that a patient with psychiatric conditions had a longer recovery period of approximately 374 days.

One of the complications encountered in our study was a septic complication characterised by surgical wound dehiscence. To manage this complication, we implemented a comprehensive treatment approach. This involved the removal of the synthetic media, curettage, and debridement of the surgical site. Additionally, an antibiotic spacer was applied to promote infection control. To provide stability and facilitate the healing process, an external fixator was placed in distraction. This combination of interventions effectively resolved the septic complication and allowed for further progress in the healing process.

Another complication observed in our study was an injury to the short flexor muscle of the fingers. Although this complication occurred, it is important to highlight that it was an isolated case and did not affect the overall success of the arthrodesis procedure.

Importantly, none of the patients experienced pain near the apex of the retrograde nail, indicating that the surgical technique used was successful in preventing pain-related complications. Additionally, no signs of radiolucency were observed, suggesting good integration and stability of the retrograde nail within the bone.

Our study found no statistically significant differences between the consolidation rates and the length of the nails used in the procedures, indicating that the choice of nail length did not have a significant impact on the fusion and overall consolidation of the ankle joint. There was no statistically significant correlation between the healing time between the smoking and non-smoking patients examined in our study (Table 2).

Overall, the results of our study demonstrate that retrograde nail arthrodesis is an effective treatment option for severe ankle fractures, leading to successful bone callus consolidation, consecutive ankle arthrodesis, and functional recovery. While complications were encountered, they have managed appropriately, and the overall outcomes were favourable. These findings support the use of this technique in cases where limb salvage is crucial and amputation is the alternative. (Figure 1)

## 4. Discussion

The goal of every traumatologist is the anatomical reconstruction of the joint surface and early mobilisation, which is a prerequisite for functional healing of the compromised joint. At the level of the leg segment and especially the ankle, these objectives cannot disregard the global assessment of the patient, thus systemic and local factors. The local factors include the state of the impairment of the local soft tissues such as skin, subcutis, muscles, tendons, vessels, nerves and above all microcirculation, which are often damaged in direct proportion to the traumatic energy responsible for the injury. Moreover, skeletal injuries are often only one aspect of a much more complex picture, such as that of a polytrauma patient, who often undergoes several operations in other sites at different times and in the manner suggested by the principles of damage control.

As we have said, the success of treatment is influenced by a number of factors, systemic and local. Systemic factors include general conditions, comorbidities, including diabetes, smoking [12], alcoholism [13], obesity, or the patient’s own poor compliance. All of these lead to a risk of non-consolidation or failure.

The cases we selected to undergo limb salvage procedures were the outcomes of failed previous surgical treatments performed in an attempt to preserve not only the limb but also its function. Therefore, an appropriate osteosynthesis was performed, often with plates, and sometimes with a circular or hybrid external fixator.

Richer et al. [14] demonstrated that the fusion rate of septic ankle joint arthrodesis with an external fixator is 62%. The remaining 39% of patients undergo revision due to secondary complications such as delayed wound healing, non-union, and infection of the external fixator. Richter et al. [15] reported that the union rate of arthrodesis was 86% in patients using a combination of internal fixation and hybrid external fixation, justifying that the external fixator protects against torsional rotation, but the main stability is provided by internal fixation. Klouche et al. [16] reported fusion in 89.5% of patients with ‘one-stage’ internal fixation. Simoni et al. [17] reported fusion rates of 91% for patients treated with ‘two-stage’ surgical treatment.

Tibio-calcaneal arthrodesis with a retrograde nail is not a procedure without complications as described in the literature between 21 and 60% [18]. It is necessary to empathise with the patient and establish a trusting relationship, as it is often the case that patients under 50 years of age have to overcome qualms about giving up the use of the tibiotarsal joint. Patients must understand that in special situations, such as in the case of severe impairment of the articular surface, bone loss, infection, post-traumatic stiffness, tendon or neuro-vascular and skin damage, the only achievable goal is to save the limb at the expense of ankle function.

In our study, we used the RIA taken from the ipsilateral femur as the bone graft. This did not allow us to perform the compression of the fragments. Therefore, the consolidation time was longer than the data in the literature referring to standard arthrodesis without bone loss. Another complication encountered was the injury of the short flexor muscle of the toes. This can be explained via the plantar introduction of the nail to obviate this risk; the safe zone seems to be identified in the joint between sustentaculm-tali and calcaneus [19]. We found no signs of radiolucency or tibial pain at the apex of the nail, as reported in the literature [20]. This is because we stabilised the nail with two proximal and three distal screws. Concerning the number of screws, although it is difficult to match the positioning of the screws exactly in the body of the talus and the calcaneus due to anatomical variability, we always paid attention that the distal screw on the calcaneus is always present and with posterior-anterior direction, as it is, according to the literature, the main stabiliser in the anti-rotatory sense [21].

## 5. Conclusions

In conclusion, the retrograde nailing of the tibio-astragalic joint using autologous bone grafts from the ipsilateral femur, along with the RIA technique, proved to be a valuable salvage procedure in selected cases of severe joint impairment and local tissue damage. This approach offers a promising alternative to amputation in situations where limb preservation is essential.

Our study shows that two-stage RIA-assisted ankle arthrodesis can be applied in cases of limb salvage with post-traumatic infectious complications because all patients achieved bone callus consolidation, ankle arthrodesis and functional recovery. The average consolidation time of seven months indicates a relatively short period for achieving joint fusion. Additionally, patients were able to resume their daily work activities within an average of eight months post-surgery, indicating a satisfactory recovery period. The resolution of the initial limp during walking occurred at approximately one year, with the exception of a single patient with psychiatric conditions.

Importantly, the complication rates were low in our study. We encountered only one septic complication, which was effectively managed via a comprehensive treatment approach. No reports of radiolucency or tibial pain at the apex of the nail were observed, suggesting the excellent integration and stability of the retrograde nail within the bone. These findings indicate the effectiveness of the surgical technique in preventing postoperative complications and promoting successful outcomes.

Retrograde nail arthrodesis should be considered as a viable standardised protocol treatment option in cases where limb salvage is of utmost importance. It offers the potential for achieving joint fusion, preserving functionality, and avoiding amputation in severe cases involving joint impairment and local tissue damage. The use of autologous bone grafts from the ipsilateral femur, along with the RIA technique, enhances the success of the procedure.

However, further research and long-term follow-up studies are necessary to validate our findings and evaluate the long-term effectiveness of this technique in a larger patient population. Additional investigations should focus on assessing functional outcomes, evaluating patient satisfaction, and comparing this technique with other treatment modalities. Moreover, the examination of potential factors that may affect the success of retrograde nail arthrodesis, such as patient characteristics and the presence of comorbidities, would provide valuable insights for optimising patient selection and improving overall outcomes.

In conclusion, retrograde nail arthrodesis with autologous bone grafts using the RIA technique represents a promising salvage method in severe cases of joint impairment and local tissue damage. Its success in achieving bone consolidation, ankle arthrodesis, and functional recovery, coupled with low complication rates, warrants its consideration as a treatment option for limb salvage. Continued research and long-term studies will further establish the effectiveness and long-term outcomes of this technique, benefiting a broader population of patients in need of such interventions.

## Figures and Tables

**Figure 1 jfmk-08-00122-f001:**
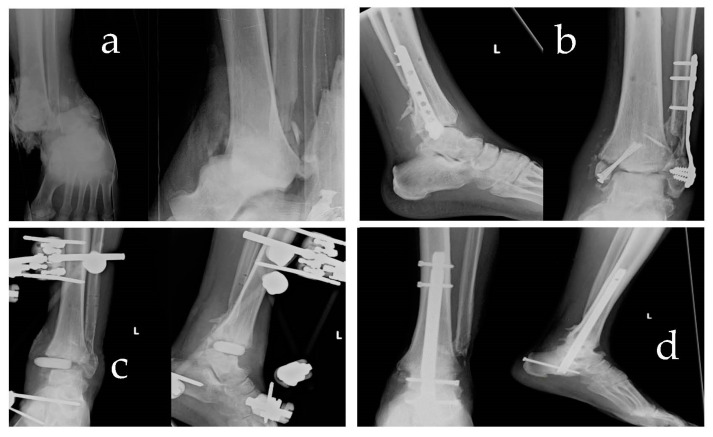
Figure shows a clinical case examined. The image in (**a**) shows fracture dislocation type 44B3. The image in (**b**) shows clinical follow-up at 2 months with media exposure and staphylococcus aureus mrsa infection. (**c**) Control post debridement and curettage of devitalised tissue, removal of synthesis devices, stabilisation with external fixator, and implant of antibiotic spacer. (**d**) Clinical follow-up at 6 months of arthrodesis with retrograde nail and RIA graft.

**Table 1 jfmk-08-00122-t001:** Graphically simplifies the characteristics of the sample under examination.

Patients	35
Male/Female	23 (65.7%) Male–12 (34.3%) Female
Age	47.8 ± 20.08 (22–85)
AO/OTA	18 (51.4%) pts 43 C3 10 (28.5%) pts 44 C 7 (20%) pts 44 B
Infections	16 (45.7%) pts *Staphylococcus aureus* 12 (34.2%) pts *Staphylococcus aureus* MRSA 3 (8.5%) pts *Escherichia coli* 4 (11.4%) pts *Acinetobacter baumani*
Comorbidities	2 (5.7%) pts diabetes mellitus tp II+ hypertensions 5 (14.2%) pts hypertensions 23 (65.7%) pts smokers 5 (14.2%) pts no other comorbidities

**Table 2 jfmk-08-00122-t002:** Describes the results obtained.

	Mean Value	*p*-Value
Healing (days)/nail length (mm)	229.4/197.1	0.107661
Healing (days)/smoker–no smokers (n pts)	229.4/23 229.4/12	0.114005
Complications	2 (5.7%)	n/a

## Data Availability

The data presented in this study are available in the text and tables.

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
