# Peer review of "Tibiotarsal Arthrodesis with Retrograde Intramedullary Nail and RIA Graft: A Salvage Technique"

_jfmk, 2023, doi:10.3390/jfmk8030122_

Round 1

Reviewer 1 Report

Undertaking a comprehensive exploration of ankle arthrodesis as a salvage procedure, this study provides a thorough comparative analysis of techniques, emphasizing on infection control measures, potential complications, and clearly defining success measures and outcomes, while enhancing the context with existing literature and shedding light on our unique contributions in this domain. However, here are some potential improvements that could be made: 

1. For Introduction: Comparative analysis of techniques, thorough elaboration on infection control measures and potential complications, clear definition of success and outcomes, connection to existing literature, explanation of the study's contribution, and discussion on why ankle arthrodesis is considered a salvage procedure.

2. For Materials and Methods: More comprehensive study design details, extended demographic information, detailed discussion of the decision-making process for choosing surgical techniques, thorough statistical analysis, clarity on outcome measurements, discussion about complications and follow-up protocols, and inclusion of ethical approval.

3. For Results: Explicit highlighting of main findings, contextualization of results, detailed discussion on complications, reporting negative results, description of the statistical analysis, initial interpretation of findings, synthesizing conclusions, and inclusion of visual aids to summarize results.

4. For Discussion: Clear structure, explicit connections to results, comparisons with other research, thorough explanation of complex concepts, highlighting of clinical implications, discussion of limitations, and final conclusions.

5. For Conclusions: Conciseness, clarity of key takeaways, discussion of limitations, underlining clinical implications, detailing future research, avoiding overstatement of findings, and emphasizing the importance of the study.

6. For the overall study: Enhancing demographic diversity and increasing patient sample size, clarifying success measures, providing more explicit details of the procedure, inclusion of other variables such as overall health and lifestyle, discussing failed cases, and improving the description of postoperative care.

For Abstract:

This is generally clear, but it might be beneficial to include more specific information about the outcomes. Instead of stating "we present our experience," you could say "we present successful limb salvaging results..." to offer a preview of the results.

For Introduction:

Consider breaking up the larger paragraphs for easier reading. You could also clarify some of the technical language. For instance, instead of saying "The objective of this study is to report our experience," you could say "This study aims to share our successful experience with limb salvage using ankle arthrodesis in patients suffering from post-traumatic outcomes."

For Materials and Methods:

This section could benefit from more detail about the methodology used. For example, you might want to expand upon the procedures for collecting and analyzing data.

Additionally, consider replacing "laboratory analysis revealed positive results for..." with "the bacteria found to be responsible for the infections included Staphylococcus aureus, MRSA, Escherichia coli, and Acinetobacter baumani in 16, 12, 3, and 4 patients respectively."

Author Response

 I wanted to extend my heartfelt gratitude for taking the time to review and provide insightful corrections for our scientific. Your expertise and thoughtful suggestions have immensely contributed to the enhancement of the quality and rigor of our work.We truly appreciate your commitment to upholding the standards of academic excellence and your willingness to share your expertise to help us refine our work. Please rest assured that we have carefully considered each of your corrections and incorporated them into our revised manuscript. We believe that your contributions will significantly elevate the impact and credibility of our research within the academic community.

 If you have any further suggestions or would like to discuss any aspect of our article, please do not hesitate to reach out.

Reviewer 2 Report

This is a manuscript in which the authors reflect their experience. I believe it is an excellent work, with the limitations of a case study description. Even so, this type of work can often show us great advances. It is complicated, if not impossible, to perform a clinical trial in this type of pathology. Therefore, I think that we should promote this type of work more. 

The Abstract is well presented. I would encourage the authors to put more emphasis on the results. 

The Introduction is well-stated, and the problem is well-focused. 

Material and Methods: evidently a larger number would be needed for the results to be more conclusive. The surgical description is well argued.

Results: I believe that Table 1 should appear here, not in the previous section.

Discussion: there is not really a discussion, there is an account of events, and it should provoke more debate. 

Conclusions: I think the authors should be more cautious in their statements, and limit them exclusively to their sample. 

Author Response

 I wanted to express my sincere appreciation for the time and effort you invested in reviewing our scientific article, titled.Your insightful feedback and thoughtful suggestions have been invaluable in refining our work and ensuring its accuracy and impact.

We are pleased to inform you that we have carefully addressed the points you raised in your review. Your guidance has been instrumental in enhancing the quality of our manuscript, and we are pleased to share that the necessary corrections have been made according to your recommendations.

We have taken your feedback to heart and have placed more emphasis on the results in the revised manuscript. Your positive remarks about the clarity of our introduction and problem statement are encouraging and motivate us to continually refine our work to meet the highest standards.

Your recommendation to move Table 1 to the Results section has been implemented in our revised manuscript, and we believe this adjustment enhances the flow of our presentation.

Your comment about the discussion section has prompted us to reevaluate and rework that part of the article. 
Finally, your feedback on our conclusions is well-received. We understand the importance of being cautious and circumspect in our statements, and we have refined our conclusions to be exclusively relevant to our sample.

Round 2

Reviewer 1 Report

The article under discussion offers a comprehensive and thorough exploration of ankle arthrodesis as a salvage procedure. The author has not only undertaken a detailed comparative analysis of techniques, emphasizing infection control measures, potential complications, and clearly defining success measures and outcomes, but also enriched the context with existing literature, highlighting unique contributions in this domain.

The proposed improvements are so encompassing and thoughtfully considered that they have the potential to elevate the study to a higher level of rigor and relevance. These corrections, ranging from the analysis of methods and results to discussions and conclusions, show that the author is deeply committed to the integrity and quality of the research.

Recommendations for the introduction, methods, results, discussions, and conclusions are clearly articulated and focus on essential aspects for making the study both comprehensive and accessible. The proposals for the overall study, including demographic diversity and detailing success measures, are particularly impressive.

In conclusion, this article represents a significant contribution to the medical literature and should be considered for publication.